# Booster Vaccination Against Invasive Pneumococcal Disease and Hepatitis B in Previously Vaccinated Solid Organ Transplant Recipients Without Seroprotection

**DOI:** 10.3390/vaccines13121253

**Published:** 2025-12-17

**Authors:** Julie Sejerøe-Olsen, Moises Alberto Suarez-Zdunek, Thomas Helbo, Lise Bank Hornung, Charlotte Sværke Jørgensen, Kasper Rossing, Michael Perch, Allan Rasmussen, Sebastian Rask Hamm, Susanne Dam Nielsen

**Affiliations:** 1Viroimmunology Research Unit, Department of Infectious Diseases, Copenhagen University Hospital—Rigshospitalet, Blegdamsvej 9, 2100 Copenhagen, Denmark; 2Department of Virology & Microbiological Preparedness, Statens Serum Institut, 2300 Copenhagen, Denmark; 3Heart and Lung Transplant Unit, Department of Cardiology, Copenhagen University Hospital—Rigshospitalet, 2100 Copenhagen, Denmark; 4Department of Clinical Medicine, Faculty of Health and Medical Sciences, University of Copenhagen, 2100 Copenhagen, Denmark; 5Department of Transplantation and Digestive Diseases, Copenhagen University Hospital—Rigshospitalet, 2100 Copenhagen, Denmark

**Keywords:** organ transplantation, vaccination, hepatitis B, pneumococcal infections, hepatitis B vaccines, pneumococcal vaccines

## Abstract

Background: Despite pre-transplantation vaccination against invasive pneumococcal disease (IPD) and hepatitis B virus (HBV), most solid organ transplant (SOT) recipients are without post-transplantation seroprotection against IPD and HBV. We aimed to determine the seroprotection rates and changes in antibody concentrations after booster vaccination against IPD and HBV in SOT recipients without post-transplantation seroprotection after pre-transplantation vaccination. Furthermore, we aimed to identify risk factors associated with non-response to booster vaccination. Methods: In this prospective cohort study, we included adult SOT recipients without post-transplantation seroprotection against IPD who then received the 23-valent pneumococcal polysaccharide vaccine (PPSV23) booster, as well as adult SOT recipients without seroprotection against HBV who then received the Engerix-B^®^ booster after pre-transplantation vaccination. Logistic regression models were used to analyze risk factors for non-response to booster vaccination. Results: We included 50 SOT recipients in analyses of booster vaccination against IPD and 52 SOT recipients in analyses of booster vaccination against HBV. Seroprotection rates were 52% after booster vaccination against IPD and 7.7% after booster vaccination against HBV. The median geometric mean concentration of pneumococcal antibodies increased from 0.54 µg/mL IgG (interquartile range, IQR: 0.35–0.77) to 1.21 µg/mL IgG (IQR: 0.87–1.62) after booster vaccination (*p* < 0.001). Having pre-transplantation seroprotection against IPD at time of listing was associated with lower odds of non-response to booster vaccination. We were not able to identify risk factors for non-response to HBV booster vaccination. Conclusions: Booster vaccination improved seroprotection against IPD, but not HBV. Further studies are needed to examine optimal vaccination strategies for SOT recipients.

## 1. Introduction

Solid organ transplantation is a lifesaving procedure for patients with end-stage organ disease [1]. To prevent graft rejection, solid organ transplant (SOT) recipients receive immunosuppressive therapy which, in turn, increases susceptibility to infections and the risk of more severe disease when infected [2]. Correspondingly, SOT recipients have a higher incidence, morbidity and mortality of invasive pneumococcal disease (IPD) [3,4], and hepatitis B infection has been associated with reduced graft survival in liver graft recipients [5,6]. These diseases are caused by infections with *Streptococcus pneumoniae* and hepatitis B virus, respectively, both of which are vaccine-preventable infections [3,7].

Pneumococcal polysaccharide vaccines (PPSV) and the 13-valent pneumococcal conjugate vaccine (PCV13) have shown a vaccine efficacy of approximately 75% against vaccine-type IPD [8,9]. Engerix-B^®^ (GlaxoSmithKline, London, UK) and Twinrix^®^ (GlaxoSmithKline, London, UK) are recombinant vaccines that have an estimated vaccine efficacy and effectiveness against hepatitis B of >90% in a healthy population [10,11,12]. However, immunocompromised patients, including SOT recipients, have impaired immune responses to most vaccines, including standard vaccination series against IPD and hepatitis B [13,14,15,16,17]. Therefore, current vaccination guidelines recommend pre-transplantation vaccination [12,18]. However, in a recent study of 136 SOT recipients, we found that despite pre-transplantation vaccination, only 43% had post-transplantation seroprotection against IPD, and only 28% SOT recipients had post-transplantation seroprotection against hepatitis B [19], leaving most SOT recipients with suboptimal protection against IPD and hepatitis B.

Little is known about the optimal strategy to induce seroprotection in SOT recipients who do not develop seroprotection following routine pre-transplantation vaccination. While there is a lack of studies on booster vaccination against IPD in adult SOT recipients, studies on the effect of hepatitis B booster vaccination in SOT recipients without seroprotection either do not report the seroprotection rates after a single booster vaccination in SOT recipients or only include kidney transplant recipients [20,21,22].

In this study, we aimed to assess the proportion of SOT recipients with seroprotection after booster vaccination in SOT recipients who were vaccinated pre-transplantation but lacked seroprotection post-transplantation. Additionally, we aimed to determine the changes in antibody concentrations and to identify risk factors associated with non-response after booster vaccination.

## 2. Materials & Methods

### 2.1. Study Overview

In this prospective cohort study, we identified adult SOT recipients without post-transplantation seroprotection against IPD and/or hepatitis B who had completed full relevant vaccination series initiated before transplantation.

All liver, lung, heart and multi-organ transplantation candidates at Copenhagen University Hospital—Rigshospitalet, Denmark, were referred to the vaccination clinic at the Department of Infectious Diseases at the time of listing for transplantation, where a structured pre-transplantation vaccination plan was initiated as previously described [19,23]. We included SOT recipients who underwent pre-transplantation vaccination at the time of listing between February 2020 and October 2022, as vaccination against IPD and hepatitis B was guided by serology during this period. These SOT recipients were offered relevant booster vaccinations as part of the standard of care. Post-booster serology was obtained at least 4 weeks after booster vaccination.

Furthermore, we only included SOT recipients who had undergone transplantation no later than 31 October 2024 and who had completed IPD and/or hepatitis B booster vaccination followed by post-booster serology as of 1 June 2025. We excluded SOT recipients who were retransplanted before post-booster serology as well as those living outside of the Zealand and Capital Regions of Denmark as they were under clinical care at other hospitals (Figure 1).

### 2.2. Booster Vaccination Strategy

As part of clinical routine, post-transplantation seroprotection was assessed at least 3 months after transplantation at the vaccination clinic at the Department of Infectious Diseases.

SOT recipients without post-transplantation seroprotection against IPD were offered a 23-valent PPSV (PPSV23).

SOT recipients without seroprotection against hepatitis B were offered up to two booster vaccinations. Engerix-B^®^ was offered as the first booster vaccination. SOT recipients with non-response after one booster vaccination were offered Fendrix^®^ (GlaxoSmithKline, London, UK) as a second booster vaccination. Additional hepatitis B serology was obtained at least 4 weeks after the second booster vaccine.

In this study, information about vaccinations was retrieved from the Danish Vaccination Registry (DDV), which includes all vaccines administered in Denmark since 2015 [24].

### 2.3. Serology

Pneumococcal antibody concentrations were quantified at the national reference laboratory for pneumococcal antibody concentrations (Statens Serum Institut, SSI, Copenhagen, Denmark) using an in-house Luminex assay. Serotype-specific test results above the range of detectability or above 50 µg/mL IgG were registered as 50 µg/mL IgG, following SSI recommendations.

Hepatitis B surface antibody (anti-HBs) concentrations were analyzed at local hospital laboratories and collected from electronic medical records.

For post-transplantation serology, we used the earliest available serology collected after completion of vaccination series and at least 3 months after transplantation.

For post-booster serology, we used the earliest available serology collected at least 4 weeks after booster vaccination and before additional booster vaccination if multiple boosters were administered.

### 2.4. Definitions

A completed vaccination series against IPD was defined as having received both PCV13 and PPSV23 or as having received a 20-valent pneumococcal conjugate vaccine (PCV20).

Seroprotection against IPD was defined as a geometric mean concentration (GMC) of ≥1 µg/mL IgG antibodies against 12 representative serotypes (1, 3, 4, 5, 6B, 7F, 9V, 14, 18C, 19A, 19F and 23F) following national clinical recommendations and previous studies [19,25].

A completed hepatitis B vaccination series was defined as a series of 3 doses of 20 µg hepatitis B surface antigen.

Seroprotection against hepatitis B was defined as an anti-HBs concentration >10 IU/L [26].

Pre-transplantation vaccination was defined as a vaccination series initiated before the date of transplantation but not necessarily completed before the date of transplantation.

Booster vaccination was defined as the first vaccine administered after both completion of a vaccination series and transplantation. Non-response was defined as not developing seroprotection after booster vaccination.

Diabetes at booster vaccination was defined as having a diabetes mellitus diagnosis registered in electronic medical records at the time of booster vaccination. Temporary forms of diabetes mellitus, such as steroid-induced diabetes mellitus, were not included as diabetes in this study.

Non-basal cell carcinoma (BCC) cancer at booster vaccination was defined as having a cancer diagnosis other than basal cell carcinoma registered in electronic medical records at the time of booster vaccination.

### 2.5. Statistical Analysis

Non-normally distributed continuous data were presented with medians and interquartile ranges (IQR) and compared with Mann–Whitney U tests. Categorical data were presented as percentages and were compared with χ^2^ tests or Fisher’s exact tests for non-paired data, as appropriate.

To assess for selection bias, these descriptive statistics were also used to compare the characteristics of included participants and excluded SOT recipients.

Seroprotection rates before and after booster vaccination were reported as percentages and compared using McNemar’s test.

Pneumococcal and anti-HBs antibody concentrations before and after booster were compared using Wilcoxon signed-rank tests.

Changes in pneumococcal concentrations after booster vaccination were log_2_ transformed and visualized in a heat map showing relative changes. Changes in anti-HBs concentrations after booster vaccination were log_2_ transformed and visualized in charts showing relative changes. To accommodate log_2_ transformation, anti-HBs concentrations <0.1 were set to 0.1.

For seroprotection against IPD, logistic regression models were used to investigate risk factors associated with non-response after booster. Each risk factor was tested before and after adjustment for age and whether vaccination was completed before or after transplantation.

A sensitivity analysis was conducted investigating the seroprotection rates using GMC ≥ 0.35 µg/mL IgG instead of ≥1 µg/mL IgG as cut-off for seroprotection against IPD.

Furthermore, as a sensitivity analysis, the models were repeated using Firth’s penalized logistic regression to improve parameter stability in small sample sizes.

In another sensitivity analysis, we additionally adjusted for the time between booster vaccination and post-booster serology to adjust for antibody decay.

In a further sensitivity analysis, logistic regression interaction models were carried out. The variables age, sex, vaccination status, serology at initial pre-transplant vaccination and diabetes mellitus at booster were all tested individually for interaction with the following three time variables: time between transplantation and booster, time between booster and post-booster serology, and time between completion of vaccination series and booster.

In an exploratory analysis, the association between risk factors and relative anti-HBs concentration changes was examined using linear regression models. Each risk factor was tested before and after adjustment for age and whether vaccination was completed before or after transplantation.

In a sensitivity analysis, using Spearman correlations, we additionally examined the correlation between pre-transplant pneumococcal and anti-HBs concentrations obtained at initial vaccination and the corresponding antibody concentrations after booster vaccination.

All analyses were conducted in R version 4.3.2 (R Foundation for Statistical Computing, Vienna, Austria).

## 3. Results

### 3.1. Cohort Characteristics

We identified 61 SOT recipients who were vaccinated before transplantation and did not have post-transplantation seroprotection against IPD and thus required booster vaccination. Of these, 54 received a PPSV23 booster, and post-booster serology was available for 51. One SOT recipient was excluded due to re-transplantation before post-booster serology, resulting in 50 SOT recipients included in analyses of IPD seroprotection (Figure 2A).

We identified 70 SOT recipients who were vaccinated before transplantation and did not have post-transplantation seroprotection against hepatitis B and thus required booster vaccination. Of these, 59 received an Engerix-B^®^ booster, and post-booster serology was available for 54. Two SOT recipients were excluded due to re-transplantation before post-booster serology, resulting in 52 SOT recipients included in analyses of hepatitis B seroprotection (Figure 2B). Additionally, 42 SOT recipients with non-response after the first booster received Fendrix^®^ as a second hepatitis B booster vaccination, 34 of whom had serology available at least 4 weeks after the second booster.

The primary reason for exclusion was geographical location, as these SOT recipients were not followed at our outpatient center. Comparison of included SOT recipients with excluded SOT recipients showed no significant differences in age, sex or transplant type between the populations.

In total, 76 SOT recipients were included in this study, 26 of whom were included in analyses of both IPD and hepatitis B seroprotection, 24 of whom were included only in analyses of IPD seroprotection and 26 of whom were included only in analyses of hepatitis B seroprotection. Clinical characteristics are shown in Table 1.

### 3.2. Invasive Pneumococcal Disease

Among the 50 SOT recipients without post-transplantation seroprotection against IPD, 26 (52%) had seroprotection after booster vaccination (*p* < 0.001, Figure 3A).

In the group with seroprotection against IPD after booster vaccination, a larger proportion were seroprotected at the time of pre-transplantation vaccination, but were not seroprotected after transplantation, compared to those without seroprotection after booster vaccination (Table 2). No other differences between the populations were observed.

Examining pneumococcal antibody concentrations, we found that the median GMC increased from 0.55 µg/mL IgG (interquartile range, IQR: 0.36–0.78) before booster vaccination to 1.26 µg/mL IgG (IQR: 0.78–1.69) after booster vaccination (*p* <0.001) (Figure 4). Examining each serotype individually, there was a significant increase in median concentrations of all 12 serotypes after booster vaccination (Appendix A: Median pneumococcal IgG concentrations before and after booster vaccination). Pneumococcal relative changes after booster were generally positive across all serotypes, with serotype 18C having the weakest response among the 12 serotypes (Figure 5).

In a sensitivity analysis with a seroprotection cut-off for GMC of 0.35 µg/mL IgG instead of 1 µg/mL IgG, we found 10 SOT recipients to be without seroprotection at post-transplantation, 7 (70%) of whom developed seroprotection after booster vaccination (*p* = 0.02).

In logistic regression models, we found that having seroprotection at the time of pre-transplantation vaccination was associated with lower odds of non-response after booster (odds ratio [OR] 0.10 [95% confidence interval, CI: 0.01–0.60], *p* = 0.04) and after adjusting for age and whether vaccination was completed before or after transplantation (adjusted OR [aOR] 0.07 [95% CI: 0.003–0.49], *p* = 0.02). We did not identify other risk factors associated with non-response to pneumococcal booster vaccination (Table 3).

In a sensitivity analysis examining the Spearman correlation between pneumococcal antibody concentrations at initial pre-transplant vaccination and antibody concentrations after booster vaccination, we found a moderate correlation (rho = 0.322, *p* = 0.02).

In sensitivity analyses using Firth’s penalized logistic regression, all results remained consistent with the primary analyses.

In a sensitivity analysis adjusting for antibody decay, the addition of time between booster vaccination and post-booster serology did not affect the primary analyses.

In a sensitivity analysis testing interaction models between baseline characteristics and time intervals, no significant interactions were found.

### 3.3. Hepatitis B

Among 52 SOT recipients without post-transplantation seroprotection against hepatitis B, 4 (7.7%) obtained seroprotection after booster vaccination (*p* = 0.13, Figure 3B).

The median anti-HBs concentration before booster was 0.35 IU/L (IQR: 0.00–0.70) compared to 0.30 IU/L (IQR: 0.00–1.20) after booster. There were no statistically significant changes (*p* = 0.21). Relative changes in anti-HBs concentrations are visualized in Appendix A.

Due to the low number of SOT recipients with seroprotection against hepatitis B after booster vaccination, it was not possible to assess risk factors for non-response using logistic regression models. In an explorative analysis of relative anti-HBs concentration changes, no significant associations were found (Appendix A).

Among the 34 SOT recipients who received a second hepatitis B booster vaccination, 4 (12%) obtained seroprotection (*p* = 0.13, Figure 3B).

The median anti-HBs concentration increased from 0.13 IU/L (IQR: 0.00–0.54) before the second booster to 0.61 IU/L (IQR: 0.00–2.50) after the second booster (*p* < 0.001). Relative anti-HBs concentration changes are visualized in Appendix A.

In a sensitivity analysis examining the Spearman correlation between anti-HBs concentrations at initial pre-transplant vaccination and anti-HBs concentrations after the first booster vaccination, no correlation was found (rho = −0.015; *p* = 0.92).

## 4. Discussion

In this cohort study of adult SOT recipients who were vaccinated before transplantation but did not obtain post-transplantation seroprotection against IPD, we found that the seroprotection rate against IPD following booster vaccination was 52%. Additionally, pneumococcal antibody concentrations were significantly higher after booster vaccination. For SOT recipients who were vaccinated before transplantation but did not obtain post-transplantation seroprotection against hepatitis B, we found a seroprotection rate of 7.7% after the first booster vaccination. Among SOT recipients who received a second HBV booster, we found a seroprotection rate of 12%.

We found a seroprotection rate of 52% after booster vaccination against IPD. A ≥0.35 µg/mL IgG cut-off for seroprotection against IPD is commonly used based on recommendations from the World Health Organization as a measure of serotype-specific protection against IPD [26]. In this study, antibodies were determined by the Danish reference laboratory SSI using Luminex. We used a GMC ≥ 1 µg/mL as a general cut-off across 12 representative serotypes, as this is the GMC cut-off that using Luminex technology most closely resembled a serotype-specific IgG ≥ 0.35 µg/mL across all serotypes [27]. Furthermore, it has been shown that the different pneumococcal serotypes have varying thresholds for seroprotection against IPD, suggesting that GMC ≥ 0.35 µg/mL IgG would be an underestimation [25]. In a sensitivity analysis using a GMC ≥ 0.35 µg/mL IgG cut-off, the seroprotection rate was higher at post-booster serology compared to pre-booster serology, which is consistent with our primary analysis.

We found that having seroprotection at pre-transplantation vaccination was associated with lower odds of non-response, suggesting some immunological memory despite the lack of post-transplantation seroprotection. We did not identify other risk factors, which is in accordance with our previous study [19]. These findings remained consistent through all sensitivity analyses. Considering the small sample size, and the possibility that antibody concentration measurements may fluctuate around the threshold for seropositivity, our results should be interpreted with caution.

In this study, all SOT recipients received PPSV23 as booster vaccination against IPD, but other IPD vaccinations could have been used as booster vaccinations. In contrast to the T-cell independent response delivered by PPSV, PCV generates a T-cell dependent response with immunological memory and high-affinity antibodies [18,28]. Therefore, it is possible that PCV booster vaccination could result in a higher seroprotection rate after a booster. However, several types of maintenance immunosuppressive medication target T-cell activation or proliferation [29], which could mitigate the effect of booster vaccination with PCV in SOT recipients. Future studies could compare PPSV and PCV booster vaccination in SOT recipients.

A seroprotection rate of 7.7% against hepatitis B after booster vaccination is low compared to other studies. A study of 40 heart and lung transplant recipients found a seroprotection rate of 37.5% after booster vaccinations with Heplisav-B^®^ or Recombivax^®^ [21]. Similarly to Engerix-B^®^ and Fendrix^®^, Heplisav-B^®^ and Recombivax^®^ are recombinant vaccines containing hepatitis B surface antigen, but the four vaccine types contain different adjuvants. Heplisav-B^®^ was administered in two 20 µg booster doses, which is its standard series, and Recombivax^®^ was administered in three 40 µg booster doses, which is quadruple-strength compared to its standard series and normally used for dialysis patients. Therefore, this study examined the effect of an entire booster series instead of a single booster, which is likely to explain the higher seroprotection rate. Another study of 17 kidney transplant recipients without serological or cellular immunity examined the effect of Fendrix^®^ booster vaccination and found that 7 (41%) obtained seroprotection [22]. However, prior to booster vaccination, these kidney transplant recipients had received heterogeneous vaccination series, including variability in number of vaccinations and dosage of vaccinations, making it difficult to compare with our results. To obtain seroprotection, it is possible that other booster strategies should be examined, such as higher dosage or an increased number of doses [21,30]. Furthermore, it is possible that some vaccine types elicit a higher seroprotection rate in SOT recipients compared to Engerix-B^®^ or that timing of booster vaccination can be optimized. Future studies should examine the effect of these alternative strategies.

While we found a low seroprotection rate, this does not preclude a hepatitis B booster vaccination effect, as we did find a slight increase in anti-HBs concentrations following a second booster. An anti-HBs >10 IU/L 1–2 months after hepatitis B vaccination in the general population is estimated to be protective against hepatitis B for over 30 years despite a subsequent decline in anti-HBs concentration below this threshold [31,32]. Although it is unclear to what degree this could be extended to SOT recipients, it does indicate that immunity can be present despite antibody concentrations below threshold. Furthermore, we did not examine whether the included SOT recipients developed cellular immunity. The presence of cellular immunity has been demonstrated in kidney transplant recipients without antibody response after vaccination [33]. Therefore, it is possible that hepatitis B vaccination and booster vaccination still amounts to a degree of protection despite the lack of antibody seroprotection. However, no markers of cellular immunity or surrogates were available in our study, and a potential effect on cellular immunity remains speculative.

In this study, we found a large discrepancy in seroprotection rates between SOT recipients receiving vaccinations against IPD and hepatitis B, respectively. In contrast to PPSV23 vaccination, which elicits a T-cell independent response, hepatitis B vaccination elicits a response that is dependent on T-cells [34]. A large proportion of included SOT recipients receive calcineurin inhibitors, which are known to have a direct inhibitory effect on T-cells [35]. It is therefore possible that immunosuppressants received by SOT recipients affect these vaccinations disproportionally, resulting in a more impaired vaccine response against hepatitis B compared to IPD.

In an explorative analysis testing for factors associated with relative anti-HBs changes, no associations could be demonstrated. However, due to the few hepatitis B responders and the size of the study population, immunosuppressive medications could not be tested in neither the IPD nor hepatitis B subsets. The impact of specific immunosuppressants on vaccination response should be examined in future studies.

The strengths of this study included access to complete data through medical records and national databases, and that pneumococcal concentrations were analyzed at a centralized national reference laboratory. Furthermore, this study reflects a clinical setting and contributes with knowledge on the effect of booster vaccination in a population at high risk of complications to infectious diseases.

The limitations in this study included the lack of a control group, which limited the assessment of booster effects. Therefore, we are not able to definitively conclude whether changes in antibody concentrations are due to the administered booster vaccine or other potential factors. Furthermore, this study did not explore risk factors for hepatitis B non-response, as logistic regression was not possible in the hepatitis B subset due to the low seroprotection rate after hepatitis B booster.

## 5. Conclusions

Seroprotection rates and antibody concentrations increased significantly after booster vaccination against IPD. Having seroprotection against IPD at the time of pre-transplantation vaccination was associated with lower odds of non-response to booster vaccination, suggesting immunological memory. In contrast, seroprotection rates against hepatitis B were comparable before and after booster. After booster vaccination, 48% remained without seroprotection against IPD and 92% against HBV. Further studies are needed to examine optimal vaccination strategies for SOT recipients.

## Figures and Tables

**Figure 1 vaccines-13-01253-f001:**
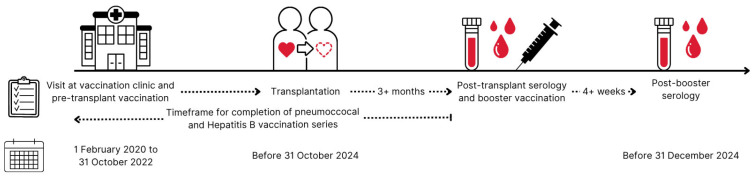
Timeline for included solid organ transplant recipients. Solid organ transplant candidates visited the out-patient vaccination clinic at Copenhagen University Hospital—Rigshospitalet for their initial vaccination at the time of listing for organ transplantation from 1 February 2020 to 31 October 2022. Solid organ transplant recipients underwent transplantation before 31 October 2024. Post-transplant serology was collected after completed vaccination series and at least 3 months after transplantation. Relevant booster vaccination was administered as soon as post-transplant serology was available. Post-booster serology was collected at least 4 weeks after booster vaccination and no later than 31 December 2024 to be included in this study. Solid organ transplant recipients who did not follow this timeline were excluded from the study.

**Figure 2 vaccines-13-01253-f002:**
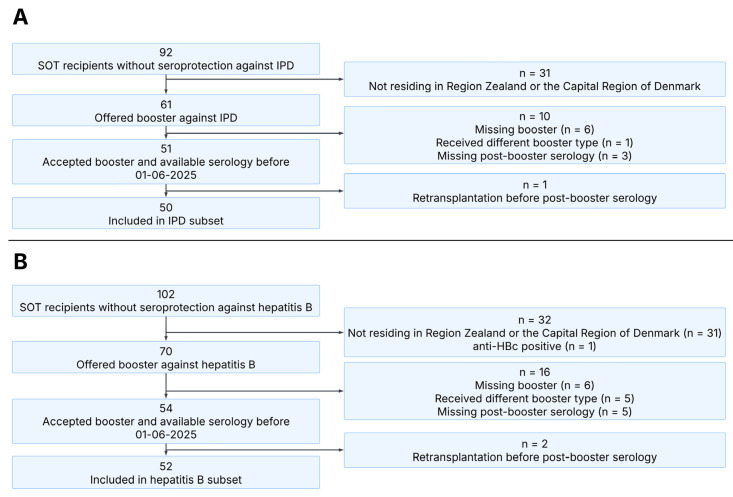
Consort diagram illustrating the inclusion of participants in this study. Panel (**A**): Overview of SOT recipients included in the IPD subset. Panel (**B**): Overview of SOT recipients included in the hepatitis B subset. SOT, solid organ transplant; IPD, invasive pneumococcal disease.

**Figure 3 vaccines-13-01253-f003:**
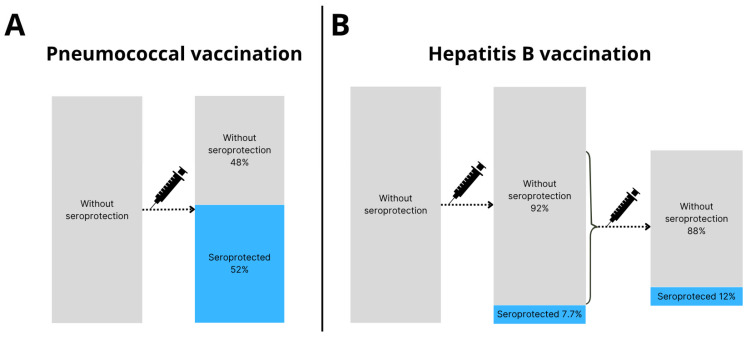
Seroprotection rates after booster vaccination. (**A**) Pneumococcal booster vaccination was administered to 50 SOT recipients, of whom 26 (52%) were seroprotected and 24 (48%) were not seroprotected >4 weeks after booster vaccination. (**B**) Hepatitis B booster vaccination was administered to 52 SOT recipients, of whom 4 (7.7%) were seroprotected and 48 (92%) were not seroprotected >4 weeks after booster vaccination. Of the 48 SOT recipients without seroprotection after HBV booster vaccination, 34 received a second booster vaccine. After the second HBV booster vaccine 4 out of 34 (12%) were seroprotected.

**Figure 4 vaccines-13-01253-f004:**
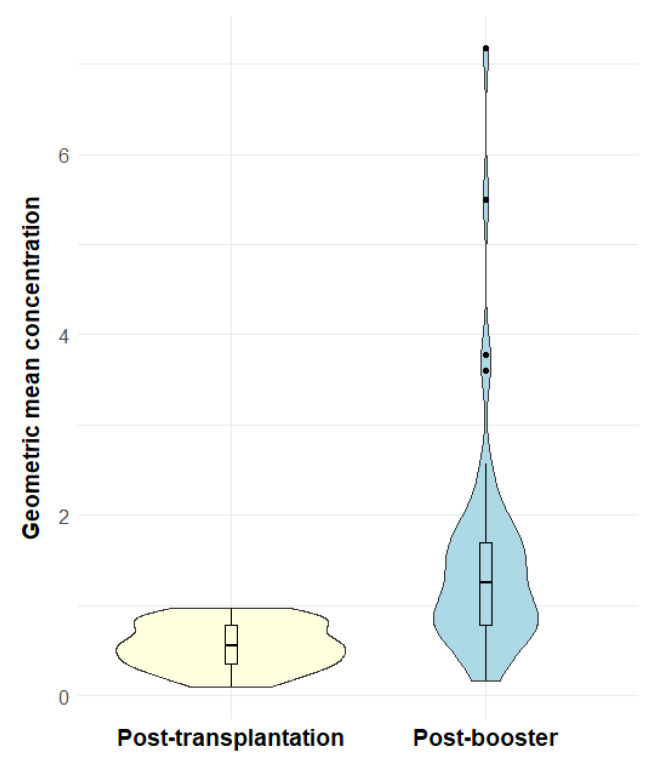
Violin plots of geometric mean concentration before and after booster vaccination. The violin plots show the probability density of geometric mean concentrations against 12 representative serotypes in µg/mL IgG of post-transplantation serology samples (yellow) and post-booster serology samples (blue). Overlaid boxplots show the distribution of the geometric mean concentrations. The boxes show the median and the interquartile ranges and the whiskers show the range of data within 1.5 interquartile range from the first and third quartile. Outliers are shown as dots outside the whiskers.

**Figure 5 vaccines-13-01253-f005:**
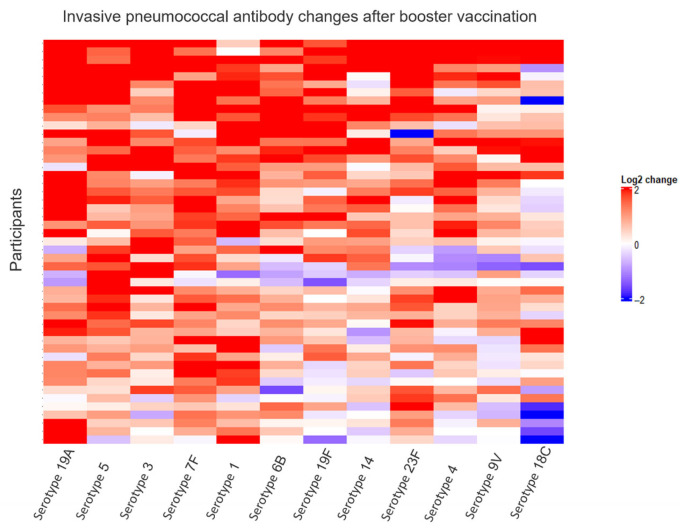
Relative changes in invasive pneumococcal antibody changes after booster vaccination. This heat map visualizes relative changes in antibody concentrations after booster vaccination. The color represents changes in pneumococcal antibody concentrations from before to after booster vaccination on a log_2_ scale, with blue representing decreases and red representing increases. Each row visualizes one participant, and each column represents one of twelve pneumococcal antibody serotypes. Columns and rows were arranged using unsupervised hierarchical clustering based on Euclidian distances and the complete linkage method.

**Table 1 vaccines-13-01253-t001:** Clinical characteristics at time of first booster vaccination. IQR, interquartile range; Non-BCC, non-basal cell carcinoma; MMF, mycophenolate mofetil.

	Invasive Pneumococcal Disease Subset (*n* = 50)	Hepatitis B Subset (*n* = 52)
**General**		
Age (years), median (IQR)	55 (47–61)	59 (46–63)
Male sex, *n* (%)	31 (61%)	26 (50%)
Organ transplant type, *n* (%)		
*Liver*	29 (58%)	30 (58%)
*Lung*	10 (20%)	15 (29%)
*Heart*	6 (12%)	3 (5.8%)
*Kidney*	3 (6.0%)	1 (1.9%)
*Dual organ transplantation*	2 (4.0%)	3 (5.8%)
Re-transplantation, *n* (%)	3 (6.0%)	5 (9.6%)
**Vaccination Information**		
Vaccination completed after transplantation, *n* (%)	14 (28%)	34 (65%)
Seroprotection at time of pre-transplantation vaccination, *n* (%)	9 (18%)	1 (1.9%)
Time from transplantation to booster (years), median (IQR)	2.24 (1.31–2.81)	1.59 (1.12–2.29)
Time between vaccination completion to booster (years), median (IQR)	2.58 (2.01–3.59)	1.36 (1.04–2.06)
Time from booster to post-booster serology (months), median (IQR)	4.92 (4.58–7.87)	5.21 (4.29–7.31)
**Comorbidity**		
Diabetes, *n* (%)	11 (22%)	7 (13%)
Non-BCC cancer, *n* (%)	3 (6.0%)	5 (9.6%)
Dialysis, *n* (%)	0 (0%)	0 (0%)
**Immunosuppressive Therapy**		
Corticosteroids, *n* (%)	35 (70%)	38 (73%)
Calcineurin inhibitors, *n* (%)		
*Tacrolimus*	43 (86%)	42 (81%)
*Ciclosporin*	6 (12%)	9 (17%)
*None*	1 (2.0%)	7 (13%)
Antimetabolites, *n* (%)		
*MMF*	43 (86%)	42 (81%)
*Azathioprine*	2 (4.0%)	3 (5.8%)
*None*	5 (10%)	7 (13%)
mTOR inhibitors, *n* (%)	4 (8.0%)	4 (7.7%)

**Table 2 vaccines-13-01253-t002:** Clinical characteristics at time of pneumococcal booster vaccination stratified by result of post-booster pneumococcal antibody serology—IQR, interquartile range; Non-BCC, non-basal cell carcinoma; MMF, mycophenolate mofetil.

Clinical Characteristics at Time of Pneumococcal Booster Vaccine	Seroprotected (*n* = 26)	Not Seroprotected (*n* = 24)	*p*
**General**			
Age (years), median (IQR)	54 (50–59)	58 (47–64)	0.482
Male sex, *n* (%)	19 (73%)	12 (50%)	0.165
Organ transplant type, *n* (%)			0.829
*Liver*	16 (62%)	13 (54%)	
*Lung*	6 (23%)	4 (17%)	
*Heart*	3 (12%)	3 (13%)	
*Kidney*	1 (3.8%)	2 (8.3%)	
*Dual organ transplantation*	0 (0%)	2 (8.3%)	
Re-transplantation, *n* (%)	2 (7.7%)	1 (4.2%)	>0.99
**Vaccination Information**			
Vaccination completed after transplantation, *n* (%)	7 (27%)	7 (29%)	>0.99
Seroprotection at time of pre-transplantation vaccination, *n* (%)	8 (31%)	1 (4.2%)	0.024
Time from transplantation to booster (years), median (IQR)	2.21 (1.51–2.96)	2.27 (1.29–2.67)	0.763
Time between vaccination completion to booster (years), median (IQR)	2.58 (1.93–3.68)	2.72 (2.35–3.44)	0.962
Time from booster to post-booster serology (months), median (IQR)	5.04 (4.60–8.19)	4.83 (4.50–7.15)	0.442
**Comorbidity**			
Diabetes, *n* (%)	5 (19%)	6 (25%)	0.738
Non-BCC cancer, *n* (%)	1 (3.8%)	2 (8.3%)	0.602
Dialysis, *n* (%)	0 (0%)	0 (0%)	>0.99
**Immunosuppressive Therapy**			
Corticosteroids, *n* (%)	17 (65%)	18 (75%)	0.545
Calcineurin inhibitors, *n* (%)			0.827
*Tacrolimus*	23 (88%)	20 (83%)	
*Ciclosporin*	3 (12%)	3 (13%)	
*None*	0 (0%)	1 (4.2%)	
Antimetabolites, *n* (%)			0.409
*MMF*	23 (88%)	20 (83%)	
*Azathioprine*	0 (0%)	2 (8.3%)	
*None*	3 (12%)	2 (8.3%)	
mTOR inhibitors, *n* (%)	2 (7.7%)	2 (8.3%)	>0.99

**Table 3 vaccines-13-01253-t003:** Risk factors associated non-response to pneumococcal booster vaccination—OR, odds ratio; aOR, adjusted odds ratio; CI, confidence interval.

	Unadjusted Model			Adjusted Model *		
	OR	95% CI	*p*	aOR	95% CI	*p*
Age (per year of age)	1.01	0.96; 1.06	0.690			
Vaccine series completed post-transplantation	1.12	0.32; 3.91	0.860			
Male sex	0.37	0.11; 1.17	0.097	0.36	0.11; 1.16	0.095
Seroprotection at time of pre-transplantation vaccination	0.10	0.01; 0.60	0.036	0.07	0.003; 0.49	0.022
Time between transplantation and booster (per year)	0.96	0.53; 1.75	0.898	0.92	0.46; 1.83	0.818
Time between booster and blood sample (per month)	0.97	0.89; 1.04	0.431	0.97	0.88; 1.04	0.433
Diabetes	1.40	0.36; 5.61	0.624	1.32	0.33; 5.53	0.692

* Adjusted for age and vaccination series completion status.

## Data Availability

The datasets analyzed during the current study are not publicly available as they contain information that could compromise participant privacy but are available from the corresponding author upon reasonable request to the extent allowed by data protection legislation.

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
