# Peer review of "Booster Vaccination Against Invasive Pneumococcal Disease and Hepatitis B in Previously Vaccinated Solid Organ Transplant Recipients Without Seroprotection"

_vaccines, 2025, doi:10.3390/vaccines13121253_

Round 1
Reviewer 1 Report
Comments and Suggestions for Authors
The authors present useful information regarding the value or lack thereof for booster immunizations for invasive pneumococcal disease (IPD) and hepatitis B virus (HBV).
The paper was incredibly well-written and contains lots of pertinent details.
I do have a few comments/questions as follows:
- The authors note that seroprotective titers obtained pre-transplant predicted response to boosters. I didn't see any analysis of which patient categories had protective titers pre-transplant. Might they be different, e.g. heart recipients versus kidney? Men versus women? From table 2 it looks like only 31/4.2% of IPD/HBV titers pre-transplant were seroprotective. Maybe it's the underlying substrate that is important rather than solid organ transplant per se. In other words - responders respond.
- From figure 2 it appears that about half of each eligible patient population elected to not receive the booster. If it had been a small percentage, say 10%, then this wouldn't bother me. But with this big number one wonders if there was something different about the populations who elected to receive the booster versus those who did not. Table 3 gives some of that info for those in the study, but not for those who elected not to receive the boosters.
- I appreciate the authors commenting about this in the discussion, but to me it is important to note that only 27/29% (IPD/HBV) of patients completed their booster series post-transplant. We don't necessarily expect patients to have protective titer until they have completed their booster series.
Congratulations again to the authors for gathering and presenting this information.
Reviewer 2 Report
Comments and Suggestions for Authors
This study is a real-world prospective cohort that includes recipients of heart, liver, lung, and other solid organ transplants. It systematically evaluates changes in seroprotection among SOT patients who previously received vaccination but failed to develop protection after transplantation, following PPSV23 or HBV booster administration. The data sources are complete, the laboratory indicators are reliable, the figures are clear, the statistical methods are appropriate, and the study addresses a clinically important area with high unmet need and limited existing evidence, giving it substantial real-world clinical relevance.
- This study lacks a non-booster control group, which limits the assessment of booster effects. It is recommended to use the existing database to construct a self-controlled longitudinal model based on each patient’s pre- and post-booster antibody trajectories to compensate for the absence of an actual control group.
- Only 7.7% achieved protection after HBV booster, but quantitative anti-HBs differences were not shown. It is recommended to use the existing raw anti-HBs values, including quantitative values below the lower limit of detection, apply extreme-value imputation, and redraw the anti-HBs distribution to allow trend analysis.
- Table 1 shows differences in baseline characteristics between the IPD and HBV groups, but no subgroup analysis was conducted. It is recommended to use the existing data to re-analyze booster responses stratified by “initial vaccination before vs. after transplantation” to eliminate potential bias.
- The seroprotection measure in Figure 3 uses only a binary indicator and does not show continuous concentration changes. It is recommended to use existing Luminex / quantitative anti-HBs data to plot paired dot plots or fold-change charts to strengthen the numerical representation of the booster effect.
- PPSV23 results only show overall GMC changes but not the individual response patterns of the 12 serotypes. It is recommended to use existing serotype data to plot a heatmap or responder clustering analysis to better explain booster behavior across serotypes.
- Baseline antibody levels are the strongest predictor, but the study did not examine correlations between “baseline antibody concentration” and booster response. It is recommended to use existing pre-transplant IgG raw values to compute Spearman correlations to avoid the information loss caused by using binary variables.
- The logistic regression includes limited variables and the sample is close to “low event” conditions. It is recommended to use existing variables to perform Firth-corrected regression to improve parameter stability, without needing additional samples.
- The 12% protection rate after the second dose of Fendrix in the HBV group is not significant, but the improvement relative to the first dose was not compared. It is recommended to calculate paired fold-changes using existing antibody values to present the true incremental effect of the second dose.
- The manuscript states that some patients had antibody results below the detection limit, preventing analysis. It is recommended to use existing laboratory reports to recode “weak positive/weak negative” qualitative categories for semi-quantitative trend analysis.
- The post-booster serology sampling interval varies, but the study did not adjust for time-difference effects on antibody decay. It is recommended to include sampling time as a covariate in regression models using existing time-point data to avoid duration-related bias.
- The explanation for non-response to HBV booster is insufficient. It is recommended to use existing immunosuppressive therapy data to perform subgroup comparisons to explore possible inhibitory factors without additional sampling.
- The violin plot in Figure 4 shows truncated extreme values, which may shift the GMC. It is recommended to perform a sensitivity analysis using existing raw plate readings to confirm that the conclusions do not depend on truncation.
- The HBV group did not evaluate cellular immunity, yet the Discussion emphasizes its potential importance. It is recommended to supplement the Discussion by using existing blood-cell parameters to explore surrogate markers of cellular immunity.
- The small sample size prevents some variables from being included in the model. Therefore, it is recommended to use the existing full cohort (IPD + HBV) for pooled exploratory analysis to increase statistical power and support trend-level conclusions.
- The study did not discuss whether the “time interval from initial vaccination to booster” influences the booster effect. It is recommended to analyze interaction terms using the existing time-interval variables.
- The explanation for HBV booster failure in the Discussion is overly brief, and additional discussion is recommended.
Round 2
Reviewer 2 Report
Comments and Suggestions for Authors
No other suggestions.